# Tiny Lungs, Big Differences: Navigating the Varied COVID-19 Landscape in Neonates vs. Infants via Biomarkers and Lung Ultrasound

**DOI:** 10.3390/biomedicines12020425

**Published:** 2024-02-13

**Authors:** Emil Robert Stoicescu, Roxana Iacob, Emil Radu Iacob, Laura Andreea Ghenciu, Cristian Oancea, Diana Luminita Manolescu

**Affiliations:** 1Department of Radiology and Medical Imaging, ‘Victor Babes’ University of Medicine and Pharmacy Timisoara, Eftimie Murgu Square No. 2, 300041 Timisoara, Romania; stoicescu.emil@umft.ro (E.R.S.); dmanolescu@umft.ro (D.L.M.); 2Research Center for Pharmaco-Toxicological Evaluations, ‘Victor Babes’ University of Medicine and Pharmacy Timisoara, Eftimie Murgu Square No. 2, 300041 Timisoara, Romania; 3Field of Applied Engineering Sciences, Specialization Statistical Methods and Techniques in Health and Clinical Research, Faculty of Mechanics, ‘Politehnica’ University Timisoara, Mihai Viteazul Boulevard No. 1, 300222 Timisoara, Romania; 4Department of Anatomy and Embriology, ‘Victor Babes’ University of Medicine and Pharmacy Timisoara, 300041 Timisoara, Romania; 5Department of Pediatric Surgery, ‘Victor Babes’ University of Medicine and Pharmacy, Eftimie Murgu Square No. 2, 300041 Timisoara, Romania; radueiacob@umft.ro; 6Department of Functional Sciences, ‘Victor Babes’ University of Medicine and Pharmacy Timisoara, Eftimie Murgu Square No. 2, 300041 Timisoara, Romania; bolintineanu.laura@umft.ro; 7Center for Research and Innovation in Precision Medicine of Respiratory Diseases (CRIPMRD), ‘Victor Babes’ University of Medicine and Pharmacy, 300041 Timisoara, Romania; oancea@umft.ro; 8Department of Pulmonology, ‘Victor Babes’ University of Medicine and Pharmacy, 300041 Timisoara, Romania

**Keywords:** inflammatory markers, biomarkers, lung disease, lung ultrasound, neonates, infants, COVID-19, SARS-CoV-2, multisystem inflammatory syndrome

## Abstract

Due to their susceptibilities, neonates and infants face unique SARS-CoV-2 challenges. This retrospective study will compare the illness course, symptoms, biomarkers, and lung damage in neonates and infants with SARS-CoV-2 infection from February 2020 to October 2023. This study was conducted at two hospitals in Timisoara, Romania, using real-time multiplex PCR to diagnose and lung ultrasonography (LUS) to assess lung involvement. Neonates had a more severe clinical presentation, an increased immune response, and greater lung involvement. Neonates had more PCR-positive tests (*p* = 0.0089) and longer hospital stays (*p* = 0.0002). In neonates, LDH, CRP, and ferritin levels were higher, indicating a stronger inflammatory response. Reduced oxygen saturation in neonates indicates respiratory dysfunction. The symptoms were varied. Infants had fever, cough, and rhinorrhea, while neonates had psychomotor agitation, acute dehydration syndrome, and candidiasis. This study emphasizes individualized care and close monitoring for neonatal SARS-CoV-2 infections. Newborn lung ultrasonography showed different variances and severity levels, emphasizing the need for targeted surveillance and therapy. Newborns have high lung ultrasound scores (LUSS), indicating significant lung involvement. Both groups had initial lung involvement, but understanding these modest differences is crucial to improving care for these vulnerable populations.

## 1. Introduction

The COVID-19 pandemic has brought forth a unique set of challenges, particularly in understanding its impact on the pediatric population [1,2]. While children, in general, have shown lower susceptibility to severe illness compared to adults, neonates and infants present a distinct subset with their own set of vulnerabilities [2,3,4]. Among these vulnerabilities, the manifestation of COVID-19 pneumonia in these age groups has prompted a focused exploration into diagnostic methodologies that offer precision and safety [5,6,7].

Neonates and infants typically display less severe symptoms of COVID-19 and have a lower occurrence of severe cases in comparison to later age groups. Typical symptoms encompass fever and respiratory problems, while certain infants may not exhibit any signs [4,8]. Transmission generally happens via close proximity, and severe consequences are infrequent [8,9]. Treatment and management entail providing appropriate care that is customized to the individual’s age and clinical state [3,10].

In this pursuit, the role of lung ultrasound has emerged as a promising avenue, showcasing its potential to serve as a non-invasive, radiation-free, and readily available imaging tool [11,12,13]. Its application in assessing lung involvement in neonates and infants with COVID-19 pneumonia provides a window into the disease’s impact on delicate pulmonary structures and the subsequent clinical implications [9,14,15].

What sets this diagnostic modality apart is its ability to capture real-time images, offering clinicians an immediate and dynamic insight into lung pathology. This is particularly crucial in pediatric cases where exposure to radiation through traditional imaging methods might pose significant concerns [9,16,17]. Moreover, the adaptability of ultrasound in various clinical settings, including resource-limited environments, adds to its allure as a diagnostic aid for pediatric COVID-19 cases [18,19].

However, while the utility of lung ultrasound appears promising, its application in neonates versus infants introduces a layer of complexity. The unique physiological differences between neonates and older infants necessitate a nuanced understanding of how lung ultrasound might differ in these age groups [5,20,21,22]. Factors such as lung development, variations in pulmonary fluid dynamics, and anatomical disparities can influence the interpretation and diagnostic accuracy of ultrasound findings [4,14,23].

Examining this differentiation is crucial in enhancing our comprehension of COVID-19 pneumonia in these specific age groups and in developing customized therapeutic approaches [6,24,25]. By delineating the comparative effectiveness of lung ultrasound in neonates versus infants, we aim to unravel the subtleties that may alter diagnostic interpretations, impact clinical decisions, and ultimately guide optimized patient care.

This research seeks to examine the distinctions in lung involvement, biomarkers, and the role of lung ultrasound as a non-radiating method for detecting COVID-19 pneumonia in newborns and young children. This study explores the complexities of SARS-CoV-2 involvement, the difficulties faced, and the possible consequences of age-related differences on the accuracy of diagnosis. Gaining a comprehensive understanding of these subtle distinctions is crucial to developing more accurate diagnostic algorithms and improving our capacity to handle pulmonary problems associated with COVID-19 in our most susceptible juvenile populations.

## 2. Materials and Methods

This article primarily focuses on analyzing the primary symptoms and biological investigation of neonates and infants who are infected with SARS-CoV-2, while also considering the use of imaging tools. This retrospective study was conducted at the Neonatology and Neonatal Intensive Care Unit (NICU) of the ‘Pius Brinzeu’ Emergency County Hospital and at the Clinic of Infectious Diseases II and the intensive care unit ‘Dr. Victor Babes’ Clinical Hospital of Infectious Diseases and Pneumophthisiology in Timisoara, Romania, from February 2020 to October 2023.

This study was conducted in accordance with the Declaration of Helsinki and obtained clearance from the Ethics Committee of both hospitals (no. 74/18 May 2020; no. 10289/25 October 2021).

This study investigated the effects of SARS-CoV-2 infection on newborns and young children, using a real-time multiplex PCR (Polymerase Chain Reaction) test for diagnosis [26]. This study meticulously watched and recorded numerous indicators and symptoms with the purpose of comprehending the wide range of consequences this virus has on these patients. The decision-making process for hospitalization relied on assessing clinical symptoms, with special emphasis placed on children under the age of one. This emphasis was due to the unique challenge faced in detecting symptoms in this age group, where expressing distress or discomfort can be particularly challenging. As a result, a nuanced approach was taken to closely monitor and analyze the evolution of cases in these pediatrics, guided by hospital protocols.

The inclusion criteria used for neonates were the following:Subjects born from mothers who tested positive for COVID-19 at birth, indicating vertical transmission.Subjects who acquired the SARS-CoV-2 infection while being hospitalized, indicating postnatal transmission.Subjects who were discharged from the hospital but developed SARS-CoV-2 infection within the first 28 days of life. All the neonates included were full-term.

The decision to hospitalize infants was determined by their fragile age, the necessity to evaluate for bacterial infection or superinfection, medical evaluation, their ability to tolerate feeding, and the adequate level of follow-up care. We included all the infants that were hospitalized for evaluation at the Clinic of Infectious Diseases II and the intensive care unit ‘Dr. Victor Babes’ Clinical Hospital of Infectious Diseases and Pneumophthisiology in Timisoara.

The criteria for exclusion were as follows:Hospitalized patients (neonates and infants) with SARS-CoV-2 infection for a duration of less than three days;Neonates and infants with preexistent chronic lung diseases such as bronchopulmonary dysplasia, immunodeficiency, cystic fibrosis, and similar conditions;The premature neonates were excluded from the analysis in order to avoid introducing bias in the interpretation of the results.Neonates and infants who do not have the consent of their parents or legal guardians.

To systematically manage and analyze the copious data collected, the team organized them within a Microsoft Office Excel table. This tabulation covered a comprehensive array of details such as gender, age, frequency of positive PCR tests, duration of hospital stay versus recovery time, manifested symptoms (including asthenic syndrome, fever, cough, rhinorrhea, vomiting, diarrhea, acute dehydration syndrome, psychomotor agitation, loss of appetite, dyspnea, and oropharyngeal candidiasis), concurrent pathologies, inflammatory markers, results from bacterial and fungal cultures, imaging examinations, and a lung affection score derived from ultrasound scans.

The asthenic syndrome in newborns is characterized by lethargy, which refers to a state of sluggishness, apathy, hypotony, drowsy dullness, or indifference. Furthermore, in infants, the condition known as asthenic syndrome is supplementary characterized by a loss of interest in novel stimuli, increased irritability, and reduced activity levels [27,28]. Psychomotor agitation, observed in neonates, was defined as heightened levels of physical restlessness and movement, linked to an inexplicable episode of continuous crying. Infants with this symptom may struggle to maintain a stationary position, frequently demonstrating heightened restlessness, episodes of elevated heart rate, or an inability to stay calm [29,30].

This exhaustive approach aimed to offer a comprehensive understanding of the multifaceted impact of SARS-CoV-2 on pediatric patients. Particular attention was devoted to elucidating the challenges and nuances of diagnosis, monitoring, and assessing the severity of symptoms among children, particularly those in the vulnerable age bracket under one years old.

The lung ultrasound assessments were conducted uniformly across all cases within the initial days (between the 2nd and 5th days) of the patients’ hospital admission. These assessments were performed by a radiologist specializing in lung ultrasounds for newborns, children, and adults. The radiologist had a minimum of four years of experience in this specific field and had their diagnoses validated by two senior radiologists with more than ten years of expertise. To conduct these examinations, two ultrasound systems were utilized: the ultrasound system Philips EPIQ 5 utilizing the L12-5 linear array probe ((5–12 MHz)), the portable General Electric Vivid IQ equipped with a linear probe (9L-RS (2.4–10.0 MHz)) and a convex probe (4C-RS (1.5–5.0 MHz)). This standardized approach ensured a consistent and comprehensive evaluation of lung conditions in pediatric patients, utilizing cutting-edge ultrasound equipment operated by experienced radiologists in the field.

Each infant or newborn admitted to the hospital underwent a comprehensive lung assessment utilizing a 12-area scoring system. This scoring system, akin to the one detailed by Mongodi et al. for COVID-19-related pneumonia in neonates (referred to as the lung ultrasound score, or LUSS), encompassed six areas on each side of the chest (two anterior, two lateral, and two posterior), demarcated by the nipple line [31]. Within each area explored, a scoring system ranging from 0 to 3 points was applied. The grading criteria were established by evaluating artifacts and determining the presence or absence of subpleural consolidation:LUSS = 0 was assigned for a normal or physiological pattern exhibiting A-lines, along with one or two B-lines per intercostal space;LUSS = 1 was an observation of more than two B-lines (referred to as sparse B-lines) per intercostal space, coupled with associated pleural abnormalities such as irregularities or thickening;LUSS = 2 was allocated for the presence of coalescent or merging B-lines, a ‘white-lung’ appearance, or small peripheral consolidations smaller than 1 cm;A maximum score of 3 points was given for substantial peripheral consolidations wider than 1 cm, whether or not they were associated with air bronchograms.

This LUSS scoring system facilitated a detailed and nuanced assessment of lung conditions, allowing for a comprehensive summary of each patient’s lung ultrasound findings. B-lines are observed as vertical lines originating from the pleural line. They occur due to the accumulation of water either in the interlobular space (sparse B-lines) or in the alveoli (confluent B-lines). A pattern of sparse B-lines suggests an outbreak of interstitial oedema, while the confluence of B-lines reveals the development of alveolar oedema [9,13,22].

We decided to divide the cohorts into five different subgroups: vertical infection—infection occurring on the first day (x = 3); infection occurring between 1 and 14 days of life (x = 9); infection occurring between 14 and 28 days of life (x = 7); infection occurring between 1 and 6 months (x = 8); and infection occurring between 7 months and 1 year (x = 15). The categorization was determined by the individuals’ age at the time of the SARS-CoV-2 infection.

The data and analyses were carefully handled using a licensed version of MedCalc^®^ Statistical Software, namely version 22.016, developed by MedCalc Software Ltd., a company located in Ostend, Belgium. The software can be accessed at https://www.medcalc.org and was last updated in 2022. 

The Shapiro–Wilk test was used to evaluate the distribution of the plotted data. The results indicate that the Shapiro–Wilk test detected a considerable departure from the normal distribution, thus requiring the utilization of non-parametric statistical techniques. Central tendency measures were computed by utilizing medians and the interquartile range [IQR] for non-parametric variables [32]. In order to determine the correlation between symptoms and lung ultrasonography scores, statistical techniques such as the Mann–Whitney U test and cross tabs were utilized to provide a clearer representation [33]. Specifically, differences between medians were highlighted using the Mann–Whitney U test, facilitating a comprehensive understanding of the association between symptoms and lung ultrasound scores.

## 3. Results

### 3.1. Demographic Data

Out of a sample size of 42 subjects, 23 were classified as infants (age between 28 days and 1 year), and the remaining 19 were categorized as newborns (age < 28 days).

Out of the 23 infants, 12 were male (52.17%), and out of the 19 neonates, 12 were male (63.16%). The comparison of these rates revealed an incidence rate difference of −0.10, with a 95% confidence interval ranging from −0.56 to 0.34. The *p*-value was found to be 0.63.

The median age of infection in the neonatal group was 12 days, with an interquartile range spanning 2.25 to 17 days. The median weight for newborns was 3060 g, with a range of 2832.5 to 3265.5 g. Out of the neonates, 63.15% (12 individuals) were delivered via cesarean section. Moreover, the mother infection was demonstrated in a number of 15 cases, accounting for 78.94% of the total. It is important to consider the possibility of a nosocomial infection or a community-acquired SARS-CoV-2 infection in the remaining newborns. Among the moms who were tested positive for SARS-CoV-2 infection, only three newborns (15.78% of all neonates analyzed) were found to have acquired the infection vertically from their mothers. Regarding the remaining individuals of the cohort, the transmission was classified as postnatal, accounting for 84.21%.

The median age for the groups of infants was 8 months, with an IQR ranging from 6 to 11 months. The weight at the center of the distribution was 8600 g, ranging from 6400 to 10,500 g.

The contrast between the two groups is illustrated through the use of tables and figures.

### 3.2. Comparison between the Evaluation of SARS-CoV-2 in Neonates and Infants

The parameters examined in the comparison between the infant and neonate groups are displayed in Table 1. These parameters include total PCR tests, PCR positive tests, hospitalization duration, hemoglobin level (g/dL), leukocytes count (×10^9^/L), lymphocytes count (×10^9^/L), neutrophiles count (×10^9^/L), thrombocytes count (×10^9^/L), procalcitonin level (ng/mL), CRP level (mg/L), ferritin level (µg/L), LDH level (U/L), AST level (U/L), ALT level (U/L), IL-6 level (pg/mL), O_2_ saturation level (%), and LUSS.

The variations in the length of hospitalization were examined using the Mann–Whitney test. Figure 1 illustrates the analysis of the duration of the hospitalization term between infants and neonates.

The figures below illustrate the differences in two crucial inflammatory markers (CRP and LDH) between infants and neonates with SARS-CoV-2 infection. Figure 2 illustrates the disparity in medians and interquartile ranges (IQR) for C-reactive protein levels in the infant and neonate groups. Conversely, Figure 3 depicts these disparities in relation to the level of LDH.

In addition, two more significant indicators used to measure the severity of SARS-CoV-2 infection are the degree of oxygen saturation and the score obtained from lung ultrasound (LUSS). Figure 4 and Figure 5 display the comparative analysis diagrams of infants and newborns.

Table 2 displays the prevailing observations concerning the signs and symptoms observed in infants and newborns. The following symptoms were considered for analysis: psychomotor agitation, asthenic syndrome, fever, cough, rhinorrhea, acute dehydration syndrome, diarrhea, vomiting, lack of appetite, dyspnea, and candidiasis.

Table 3 presents the current observations about lung ultrasound findings in infants and newborns. The analysis took into account the following findings: sparse B-lines, confluent B-lines, anomalies in the pleura, subpleural consolidation <1 cm, consolidation measuring more than 1 cm, and pleural effusion.

Table 4 displays the comparison of LUSS incidence in various areas of interest for both groups. Figure 6 displays the cumulative score for all areas of interest in both groups. The legend for all areas of interest is consistent with the elements (rows) from Table 4.

### 3.3. Comparison between Biomarkers, Signs, and Symptoms in Subgroups

The parameters examined in the comparison between the subdivided groups of neonates and infants cohort are displayed in Table 5.

Table 6 displays the prevailing observations concerning the signs and symptoms observed in the subdivided groups of neonates and infants cohort.

## 4. Discussion

Neonates face heightened vulnerability to SARS-CoV-2 infection compared to infants due to their immature immune systems, limited transfer of protective antibodies from mothers, smaller airways leading to respiratory vulnerability, increased exposure in close contact scenarios, the potential absence of typical symptoms delaying diagnosis, susceptibility in healthcare settings, and concerns regarding transmission via breastfeeding [4,6,34]. These factors collectively accentuate the susceptibility of neonates to COVID-19, necessitating tailored care approaches and heightened vigilance in their management and protection [4,6,9,35].

The provided comparison between infants (*i* = 23) and neonates (*n* = 19) across various parameters reveals significant differences in their response to SARS-CoV-2 infection. Notably, neonates showed a higher median number of total PCR tests conducted, a greater count of PCR-positive tests, and an extended duration of hospitalization compared to infants. During the COVID-19 outbreak in Romania, a regulation stipulates that two consecutive negative tests are necessary before considering the discharge of a patient. Consequently, there has been an increase in the number of PCR tests, indicating an extended duration of hospitalization for neonates and a rise in the duration during which the virus is detectable in their PCR testing.

Hemoglobin levels were notably higher in neonates, indicating potential differences in hematological response between the two groups. Regarding the hemoglobin level, it must be taken into account that the hemoglobin levels of neonates are significantly higher than those of children [36]. In terms of white blood cell counts, neonates exhibited significantly higher leukocyte, lymphocyte, and neutrophil counts, indicating a more pronounced immune response. Notably, neonates presented lower platelet counts, although not statistically significant, suggesting potential implications for clotting and bleeding concerns in this cohort [37]. Roy et al. documented a case series of 15 newborns who were born prematurely and had restricted growth due to mothers with SARS-CoV-2 infection during pregnancy. These newborns experienced bleeding within the first two days of life [37]. Moreover, their laboratory findings indicated a hyperimmune reaction, as seen by elevated levels of procalcitonin and C-reactive protein. Furthermore, their coagulation profile was significantly disrupted, with very high levels of d-dimer, normal platelet counts, and normal-to-high fibrinogen values [37].

Markers of inflammation and tissue damage (LDH, CRP, and ferritin) were notably elevated in neonates, indicating more severe inflammatory responses. Importantly, oxygen saturation levels were lower in neonates, pointing towards more compromised respiratory function. One possible explanation for this phenomenon is the multisystem inflammatory syndrome in neonates (MIS-N), which is characterized by a heightened immunological response to the illness. This heightened immune response results in elevated levels of inflammation throughout their bodies [38,39,40]. 

The lung ultrasound score (LUSS) was significantly higher in neonates, suggesting more pronounced lung involvement in this group. The combined results emphasize that neonates experience a more serious clinical manifestation and immunological reaction to SARS-CoV-2 infection compared to babies. This highlights the importance of customized care and careful monitoring for this vulnerable population [12,35]. 

The comparison between signs and symptoms exhibited by infants and neonates with SARS-CoV-2 infection reveals notable differences in their clinical presentations. Psychomotor agitation, acute dehydration syndrome, and candidiasis were significantly more prevalent in neonates, with incidence rates notably higher compared to infants. Conversely, fever, cough, and rhinorrhea were more frequently observed in infants, although not statistically significant. These findings suggest distinct clinical manifestations between the two groups, with neonates showing a higher incidence of certain severe symptoms such as psychomotor agitation, acute dehydration syndrome, and candidiasis, while infants demonstrated higher rates of more commonly recognized COVID-19 symptoms like fever, cough, and rhinorrhea. Dyspnea, although less prevalent in both groups, showed a slightly higher incidence in neonates, while asthenic syndrome, diarrhea, vomiting, and loss of appetite exhibited comparable rates between the two cohorts. The variations highlight the necessity for customized and subtle methods in diagnosing and handling SARS-CoV-2 infections in newborns and young children, taking into account the diverse and distinct symptoms found in each group [5,20,41].

The comparison of LUS findings between infants and neonates experiencing SARS-CoV-2 infection demonstrates certain disparities in the observed lung abnormalities. Notably, both infants and neonates displayed a ubiquitous presence of sparse B-lines, indicating a prevalent but relatively mild form of lung involvement in both groups. However, when examining more severe lung pathologies, such as confluent B-lines, pleural abnormalities, and subpleural consolidations smaller than 1 cm, neonates exhibited numerically higher incidences, suggesting a tendency toward more pronounced lung abnormalities compared to infants. While these differences did not reach statistical significance, the trend implies a potential inclination toward more severe lung involvement in neonates, warranting careful monitoring and assessment of respiratory conditions in this vulnerable population. Importantly, the absence of large consolidations (>1 cm) in both groups suggests a lack of extensive lung damage or severe consolidations in these cohorts. Additionally, the incidence of pleural effusion was minimal and comparable between infants and neonates. Overall, while both infants and neonates displayed prevalently sparse B-lines, the trend toward a higher incidence of certain lung abnormalities in neonates, albeit not statistically significant, warrants closer attention to respiratory complications in neonatal SARS-CoV-2 infections. These findings align with the results obtained in other research of a similar nature [9,12,24,42,43,44].

Comparing the LUSS findings in infants and neonates affected by SARS-CoV-2 in specific areas of focus shows significant variations in the distribution of lung abnormalities. Certain patterns emerge, indicating distinct regional involvement between the two groups. Notably, while the left anterior superior and left anterior inferior areas exhibited minimal involvement in infants (0.95% and 0%, respectively), neonates showed significantly higher incidences (8.19% and 9.05%, respectively). Similarly, the left lateral superior and left lateral inferior areas displayed higher involvement in neonates compared to infants, although not statistically significant. Interestingly, the right posterior superior and right posterior inferior areas demonstrated markedly higher incidences in both infants and neonates, with neonates exhibiting slightly elevated rates compared to infants. These disparities suggest potential variations in the regional distribution and severity of lung abnormalities between infants and neonates affected by SARS-CoV-2. The statistically significant differences observed in specific areas, especially the right posterior superior and right posterior inferior regions, emphasize the need for detailed regional assessments and tailored monitoring strategies in both infant and neonatal populations impacted by COVID-19. Although the posterior regions continue to be the most affected, as previously indicated by other studies [25,42,45].

Comparing the results across different infection groups reveals several noteworthy trends. Firstly, in terms of days of hospitalization, infections occurring between 14 and 28 days of life exhibit the longest duration of hospital stay, suggesting a potentially more severe course of illness during this period. Hemoglobin levels appear relatively stable across groups, although there’s a slight decrease in infections between 14 and 28 days of life, possibly indicating some degree of anemia associated with this stage. Leukocyte counts are highest in vertical infections and infections between 1 and 14 days of life, suggesting a robust immune response during the early stages, while counts decrease in later infection periods. Lymphocyte and neutrophil counts generally follow a similar pattern, reflecting the immune system’s response to infection. Interestingly, monocyte counts are relatively consistent across groups, indicating a less variable immune response in this regard. Biochemical markers such as procalcitonin and CRP exhibit elevated levels in early and late infection periods, potentially indicating ongoing inflammation. Ferritin levels peak in infections between 1 and 6 months, suggesting a significant inflammatory response during this period. Overall, these comparisons highlight the dynamic nature of immune responses and inflammatory markers across different stages of neonatal and infant infection, underscoring the importance of customized management strategies based on the specific clinical and laboratory characteristics observed within each group.

The comparison of signs and symptoms across different stages of neonatal or Infant infection highlights several notable patterns. Psychomotor agitation appears to be more prevalent in infections occurring within the first day of life (during vertical infection) and between 14 and 28 days, indicating potential neurological involvement during these periods. Furthermore, psychomotor agitation in neonates infected with SARS-CoV-2 infection may be associated with decreased maternal-fetal perfusion and adjacent hypoxia [46,47,48]. Asthenic syndrome, characterized by weakness and fatigue, is more common in infections between 7 months and 1 year, possibly reflecting the longer duration of illness and cumulative effects on energy levels. Fever shows a progressive increase from early to later infection periods, with the highest prevalence observed between 1 and 6 months and 7 months and 1 year, suggesting a more pronounced systemic inflammatory response during these stages. Respiratory symptoms such as cough and rhinorrhea exhibit varying prevalence rates across groups, with higher frequencies in infections occurring later in infancy, potentially reflecting the increased susceptibility to respiratory pathogens as infants grow older. Acute dehydration syndrome and lack of appetite are more prevalent in later infection periods, likely due to the severity and duration of the illness impacting feeding and fluid intake. Conversely, candidiasis appears to be more common in vertical infections, possibly related to maternal transmission during childbirth [46,47,48,49].

Overall, these comparisons underscore the evolving clinical presentation of neonatal and infant infections and emphasize the importance of considering age-specific manifestations in diagnosis and management.

### 4.1. Limitations

Sample size: The study’s sample size of infants and neonates might limit the generalizability of the findings, although it is still the largest one included in such a comprehensive study. An expanded group of participants could provide a deeper understanding of the observed disparities.Variability in clinical practices: Variations in healthcare practices across different settings or regions might influence the observed differences in symptomatology, diagnosis, and management.Limited data on long-term outcomes: The study’s focus on immediate clinical presentations might overlook potential long-term effects or outcomes in infants and neonates post-infection.

### 4.2. Further Directions

Expanded comparative studies: conducting larger-scale studies involving more diverse cohorts of infants and neonates could provide a broader understanding of the nuances in lung involvement and validate the trends observed in this study.Longitudinal LUS investigations: examining the evolution of LUS findings over time in infants and neonates with SARS-CoV-2 could offer insights into the progression or resolution of lung abnormalities.Comprehensive LUS protocols: developing comprehensive LUS protocols that encompass a wider array of lung pathology and standardizing the scoring systems could enhance the accuracy and reproducibility of assessments.Validation studies with complementary modalities: conducting validation studies correlating LUS findings with other imaging modalities or clinical outcomes could validate the accuracy and reliability of LUS as a diagnostic tool in this context.

Focusing on these aspects in future studies could refine the role of LUS in assessing lung involvement in infants and neonates affected by SARS-CoV-2 and other respiratory diseases, enhance its applicability, and contribute to more robust clinical practices in managing these vulnerable populations.

## 5. Conclusions

This study reveals significant distinctions in the clinical course and manifestations of SARS-CoV-2 infection between infants and neonates. Neonates demonstrate a more severe presentation, with heightened immune responses, lung involvement, and clinical symptoms compared to infants. Notably, while both groups exhibit similar early lung involvement, the specific regional disparities and severity observed in neonates warrant focused monitoring and tailored management strategies. Understanding these nuanced differences is crucial in optimizing care and vigilance for neonates and infants affected by COVID-19, highlighting the necessity for specialized approaches in diagnosis, treatment, and ongoing assessment within these vulnerable populations.

## Figures and Tables

**Figure 1 biomedicines-12-00425-f001:**
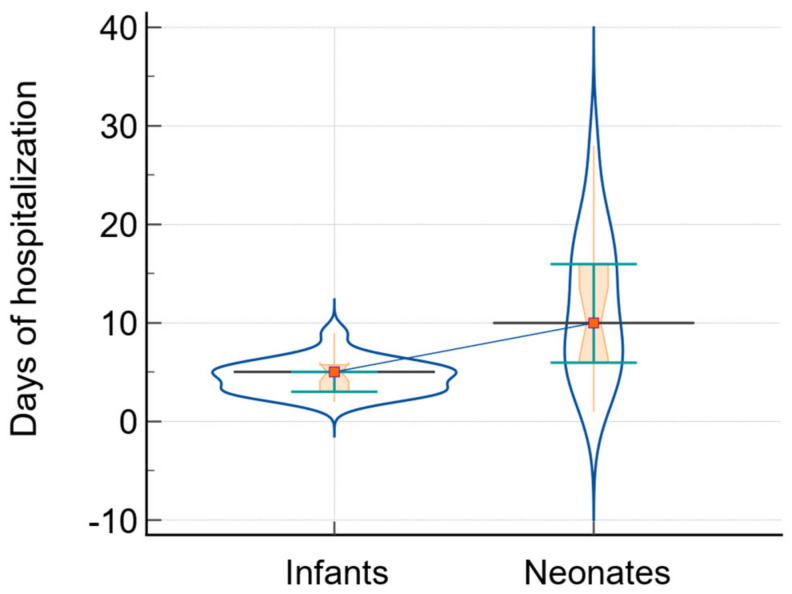
The analysis of the duration of hospitalization in infants and neonates: data comparison graph (notched box-and-whisper, violin plot with horizontal lines, markers, and connecting lines). The medians of the variables are represented by a horizontal line and markers, while the connecting line shows the relationship between them.

**Figure 2 biomedicines-12-00425-f002:**
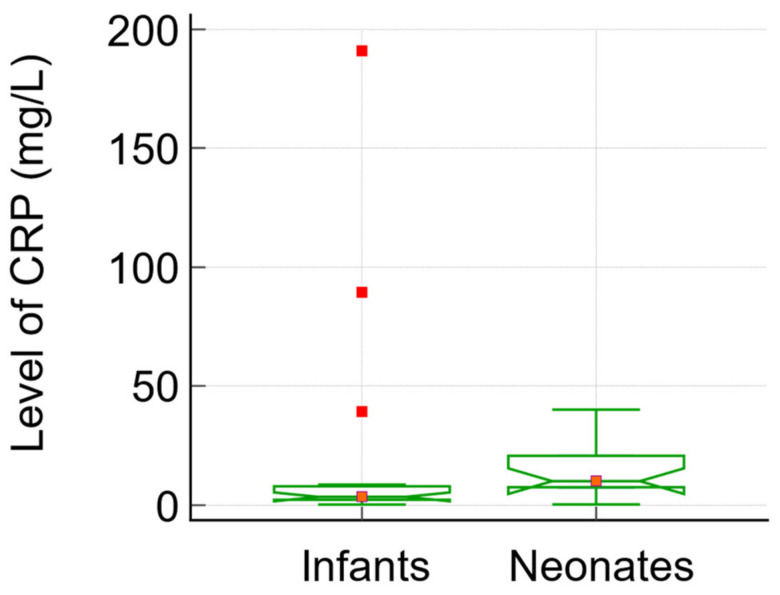
The analysis of CRP levels in infants and neonates: data comparison graph (notched box-and-whisper with markers).

**Figure 3 biomedicines-12-00425-f003:**
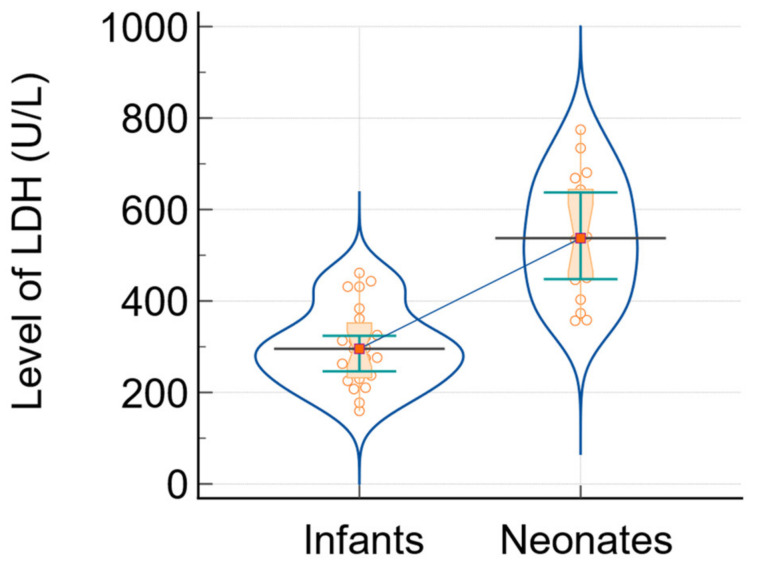
The analysis of LDH level in infants and neonates: data comparison graph (notched box-and-whisper, violin plot, dots that plot all data, horizontal lines, markers, and connecting lines). The medians of the variables are represented by a horizontal line and markers, while the connecting line shows the relationship between them.

**Figure 4 biomedicines-12-00425-f004:**
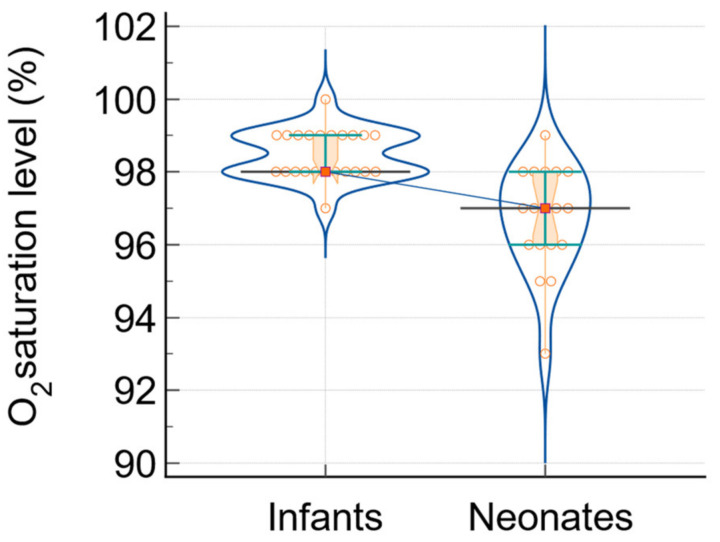
The analysis of O_2_ saturation level in infants and neonates: data comparison graph (notched box-and-whisper, violin plot, dots that plot all data, horizontal lines, markers, and connecting lines). The medians of the variables are represented by a horizontal line and markers, while the connecting line shows the relationship between them.

**Figure 5 biomedicines-12-00425-f005:**
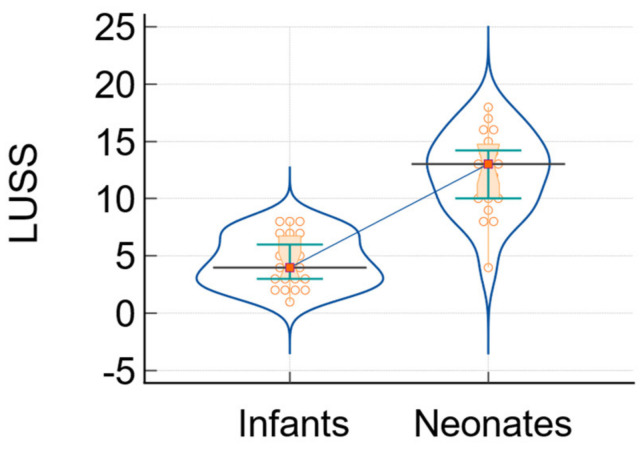
The analysis of LUSS in infants and neonates: data comparison graph (notched box-and-whisper, violin plot, dots that plot all data, horizontal lines, markers, and connecting lines). The medians of the variables are represented by a horizontal line and markers, while the connecting line shows the relationship between them.

**Figure 6 biomedicines-12-00425-f006:**
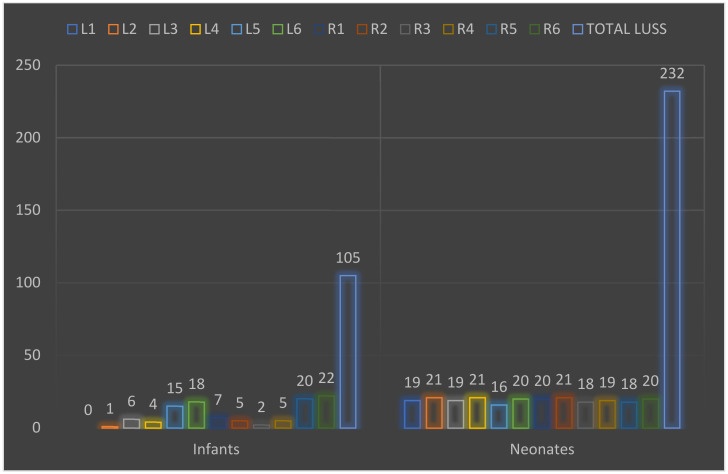
The analysis of the cumulative score for all areas of interest in infant and neonate group.

**Table 1 biomedicines-12-00425-t001:** The comparison between the determined parameters in both groups.

Parameters	Median and (IQR) for Infants (*i* = 23)	Median and (IQR) for Neonates (*n* = 19)	Mann–Whitney Test	Test Statistic Z	*p* Value
Total PCR tests	3; [3;4]	4; [3;5]	131.50	−2.36	0.0181
PCR positive tests	1; [1;2]	2; [1.25; 2]	125	−2.61	0.0089
Days of hospitalization	5; [3; 5.75]	10; [6; 16]	74	−3.67	0.0002
Hemoglobin (g/dL)	11.70; [10.92; 12.10]	14.70; [12.02; 15.87]	78.50	−3.54	0.0004
Leukocytes (×10^9^/L)	8170; [6400; 11,905]	13,200; [11,125; 20,015]	91.50	−3.20	0.0013
Lymphocytes (×10^9^/L)	3960; [2427.50; 5787.50]	5300; [4355; 7692.50]	129	n/a	0.0234
Neutrophiles (×10^9^/L)	3180; [1612.50; 4685]	5900; [2877.50; 11,020]	121	n/a	0.0131
Monocytes (×10^9^/L)	1210; [797.50; 1677.50]	1200; [885; 2595]	200	−0.46	0.6400
Thrombocytes (×10^9^/L)	345,000; [272,500; 410,750]	258,000; [218,250; 345,750]	153	−1.65	0.0978
Procalcitonin (ng/mL)	0.17; [0.10; 0.23]	0.28; [0.13; 1.02]	63.50	−1.68	0.0930
CRP (mg/L)	3.50; [2.29; 7.93]	9.63; [4.52; 12.97]	137	n/a	0.0397
Ferritin (µg/L)	121.32; [69.89; 207.80]	496.30; [282.50; 1172.50]	28	n/a	0.0001
LDH (U/L)	295; [232; 352]	540; [447.75; 651.50]	23	−4.94	<0.0001
AST (U/L)	58; [33.12; 63.40]	58; [49.25; 82.75]	167	−1.30	0.1930
ALT(U/L)	23.90; [16.47; 29.52]	26; [14.50; 32.25]	210.50	−0.20	0.8397
IL-6 (pg/mL)	8.51; [1.73; 16.35]	8.70; [4.32; 15.95]	195	−0.59	0.5526
D-dimer (mg/L)	0.78; [0.57; 1.66]	1.67; [1.20; 2.27]	111.50	−2.70	0.0068
O_2_ saturation (%)	98; [98; 99]	97; [96; 98]	52	−4.37	<0.0001
LUSS	4; [3; 6.75]	13; [10; 14.75]	15.50	−5.14	<0.0001

CRP = C-reactive protein; LDH = Lactate dehydrogenase; AST = Aspartate aminotransferase; ALT = Alanine ami-notransferase; IL-6 = Interleukin 6; n/a = not applicable

**Table 2 biomedicines-12-00425-t002:** The comparison between the incidence of signs and symptoms in both groups.

Signs and Symptoms	Number of Infants (*i* = 23) and Incidence (%)	Number of Neonates (*n* = 19) and Incidence (%)	Incidence Rate Difference	*p* Value
Psychomotor agitation	4 (17.39%)	12 (63.16%)	−0.45	0.01
Asthenic syndrome	9 (39.13%)	5 (26.32%)	0.12	0.47
Fever	19 (82.61%)	7 (36.84%)	0.36	0.06
Cough	13 (56.52%)	4 (21.05%)	0.35	0.07
Rhinorrhea	9 (39.13%)	9 (47.37%)	−0.08	0.68
Acute dehydration syndrome	19 (82.61%)	7 (36.84%)	0.45	0.06
Diarrhea	5 (21.74%)	4 (21.05%)	0.006	0.96
Vomiting	4 (17.39%)	2 (10.53%)	0.06	0.55
Lack of appetite	13 (56.52%)	10 (52.63%)	0.03	0.86
Dyspnea	1 (4.34%)	3 (15.79%)	−0.11	0.23
Candidiasis	3 (13.04%)	9 (47.37%)	−0.34	0.03

**Table 3 biomedicines-12-00425-t003:** The comparison between the incidence of LUS findings in both groups.

LUS Findings	Number of Infants (*i* = 23) and Incidence (%)	Number of Neonates (*n =* 19) and Incidence (%)	Incidence Rate Difference	*p* Value
Sparse B-lines	23 (100%)	19 (100%)	0	1.00
Confluent B-lines	7 (30.43%)	11 (57.89%)	−0.27	0.17
Pleural abnormalities	10 (43.48%)	13 (68.42%)	−0.24	0.27
Subpleural consolidation <1 cm	4 (17.39%)	6 (31.58%)	−0.14	0.55
Large consolidation >1 cm	0	0	0	1.00
Pleural effusion	1 (4.34%)	1 (5.26%)	−0.009	0.89

**Table 4 biomedicines-12-00425-t004:** The comparison between the incidence of LUSS in different area of interest for both groups.

Area of Interest	From a Total Score of LUSS (*i* = 105)	From a Total Score of LUSS (*n* = 232)	Incidence Rate Difference	*p* Value
L1—left anterior superior	0	19 (8.19%)	−0.08	0.0034
L2—left anterior inferior	1 (0.95%)	21 (9.05%)	−0.08	0.0070
L3—left lateral superior	6 (5.71%)	19 (8.19%)	−0.02	0.4397
L4—left lateral inferior	4 (3.81%)	21 (9.05%)	−0.05	0.1018
L5—left posterior superior	15 (14.29%)	16 (6.89%)	0.07	0.0383
L6—left posterior inferior	18 (17.14%)	20 (8.62%)	0.08	0.0309
R1—right anterior superior	7 (6.66%)	20 (8.62%)	−0.01	0.5573
R2—right anterior inferior	5 (4.76%)	21 (9.05%)	−0.04	0.1892
R3—right lateral superior	2 (1.90%)	18 (7.75%)	−0.05	0.0411
R4—right lateral inferior	5 (4.76%)	19 (8.19%)	−0.03	0.2748
R5—right posterior superior	20 (19.05%)	18 (7.75%)	0.11	0.0043
R6—right posterior superior	22 (20.95%)	20 (8.62%)	0.12	0.0030

**Table 5 biomedicines-12-00425-t005:** The comparison between the determined parameters in subdivided groups of neonates and infants cohort. Data are presented as median and IQR.

Parameters	Vertical Infection—Infection Occuring in the First Day (x = 3)	Infection between 1 and 14 Days of Life (x = 9)	Infection between 14 and 28 Days of Life (x = 7)	Infection between 1 and 6 Months (x = 8)	Infection between 7 Months and 1 Year (x = 15)
Days of hospitalization	6; [5.25; 7.5]	11.25; [5.50; 16]	15; [8.50; 18]	3.5; [2.50; 4.50]	5; [3.25; 6]
Hemoglobin (g/dL)	17.93; [16.90; 18.92]	15.59; [15; 16.65]	12.10; [10.85; 13.90]	11; [10.30; 11.60]	11.52; [11.12; 12.20]
Leukocytes (×10^9^/L)	12,760; [11,590; 18,580]	15,250; [11,150; 21,285]	12,100; [9980; 16,425]	8375; [7205; 9890]	7380; [6075; 12,187.50]
Lymphocytes (×10^9^/L)	4300; [3287.50; 4907.50]	5205; [4095; 6550]	6360; [4690; 8297.50]	3435; [2935; 4995]	4320; [2110; 5917.50]
Neutrophiles (×10^9^/L)	7780; [6670; 12,347.50]	6260; [4950; 13,785]	2280; [1870; 6355]	2795; [1860; 3900]	4250; [1400; 5350]
Monocytes (×10^9^/L)	1540; [835; 1907.50]	1270; [700; 3015]	1200; [925; 2430]	1235; [990; 1695]	1160; [682.50; 1637.50]
Thrombocytes (×10^9^/L)	246,000; [198,750; 251,250]	255,500; [223,500; 345,500]	320,000; [218,750; 391,250]	379,000; [331,500; 491,500]	300,000; [259,750; 358,500]
Procalcitonin (ng/mL)	0.32; [0.15; 0.82]	0.28; [0.16; 0.79]	0.16; [0.10; 5.70]	0.15; [0.10; 0.22]	0.18; [0.10; 0.26]
CRP (mg/L)	7.40; [2.22; 26.12]	8.59; [4.37; 18.66]	9.63; [5.13; 12.71]	6.02; [2.40; 23.73]	2.90; [2.28; 6.91]
Ferritin (µg/L)	216; [123; 524.25]	424.65; [216; 627]	1150; [424; 1222.50]	118.96; [71.44; 230.27]	121.32; [69.89; 188.61]
LDH (U/L)	668; [434; 718.25]	543.50; [425; 648]	540; [457.25; 651.50]	295; [217.50; 372.50]	295; [244; 322.25]
AST (U/L)	73; [55.75; 82.75]	56.50; [47.50; 72.50]	71; [52; 112]	59.30; [31.65; 69]	54.20; [33.27; 62.87]
ALT(U/L)	24; [15; 30.75]	20; [13; 28.50]	30; [25.50; 33]	25.85; [18.55; 45.35]	23.90; [15.95; 28.37]
IL-6 (pg/mL)	4.50; [2.21; 8.11]	6.07; [3.82; 12.26]	14.50; [7.26; 16.95]	10.55; [2.23; 19.43]	8.39; [1.69; 16.32]
D-dimer (mg/L)	1.67; [0.57; 2.19]	1.23; [1.13; 2]	1.89; [1.68; 2.39]	0.77; [0.68; 1.64]	0.89; [0.53; 1.62]
O_2_ saturation (%)	97; [97; 97.75]	97; [96.50; 98]	96; [95.25; 96.75]	99; [98; 99]	98; [98; 99]
LUSS	12; [9; 12.75]	12; [8.50; 13.50]	14; [10.75; 15.75]	3.50; [3; 5]	5; [2.25; 7]

**Table 6 biomedicines-12-00425-t006:** The signs and symptoms observed in the subdivided groups of neonates and infants cohort. The data are displayed in terms of value and incidence (expressed as a percentage).

Signs and Symptoms	Vertical Infection—Infection Occuring in the First Day (x = 3)	Infection between 1 and 14 Days of Life (x = 9)	Infection between 14 and 28 Days of Life (x = 7)	Infection between 1 and 6 Months (x = 8)	Infection between 7 Months and 1 Year (x = 15)
Psychomotor agitation	2 (66.66%)	5 (55.55%)	5 (71.43%)	0	4 (26.66%)
Asthenic syndrome	0	4 (44.44%)	1 (14.29%)	3 (37.50%)	6 (40%)
Fever	2 (66.66%)	1 (11.11%)	4 (57.14%)	7 (87.50%)	12 (80%)
Cough	0	1 (11.11%)	3 (42.86%)	3 (37.50%)	10 (66.66%)
Rhinorrhea	1 (33.33%)	2 (22.22%)	6 (85.71%)	2 (25%)	7 (46.67%)
Acute dehydration syndrome	1 (33.33%)	4 (44.44%)	2 (28.57%)	7 (87.50%)	12 (80%)
Diarrhea	1 (33.33%)	0	3 (42.86%)	3 (37.50%)	2 (13.33%)
Vomiting	0	2 (22.22%)	0	2 (25%)	2 (13.33%)
Lack of appetite	1 (33.33%)	5 (55.55%)	4 (57.14%)	4 (50%)	9 (60%)
Dyspnea	0	1 (11.11%)	2 (28.57%)	1 (12.50%)	0
Candidiasis	3 (100%)	4 (44.44%)	2 (28.57%)	2 (25%)	1 (6.66%)

## Data Availability

The data are encapsulated within the article. Further details can be obtained upon request from either the primary author or the corresponding author. The data are inaccessible to the public due to the patient privacy regulations governing clinical data.

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
