# Peer review of "Tiny Lungs, Big Differences: Navigating the Varied COVID-19 Landscape in Neonates vs. Infants via Biomarkers and Lung Ultrasound"

_biomedicines, 2024, doi:10.3390/biomedicines12020425_

Round 1
Reviewer 1 Report
Comments and Suggestions for Authors
The manuscript entitled “Tiny lungs, big differences: navigating the varied COVID-19 landscape in neonates vs. infants via biomarkers and lung ultrasound” has been reviewed. The authors compare some symptoms, laboratory data, and lung ultrasound between neonates and infants with COVID-19 infection. There are some questions which require further clarification.
1. The research focuses on the neonates and infants infected with SARS-CoV2. However, the authors didn’t mention how the patients were diagnosed with infection. Since Table 1 shows the comparison of PCR test numbers and positive rate, it seemed that PCR is not the final diagnosis method. It is important to explain the diagnosis method or diagnostic criteria.
2. Since this research appeared to be a multiple sites retrospective study, it is necessary to mentioned the inclusion criteria and hospitalization criteria briefly to make sure the reliability of data collected.
3. Please show the demographic data of patients collected, such as age while admitted to the hospital, gender, preterm or full term, etc. I also suggest dividing the infant group of patients into several sub-group according to their age. Some of the blood data, such as WBC and Hb are corelated to the infants’ age and other possible diseases, such as anemia.
4. The reference or descriptive criteria of some of the signs and symptoms recorded in the research, such as psychomotor agitation and asthenic syndrome, should be mentioned. Since it is a multiple sites research, we need to know the criteria of description is the same.
5. On line 175, page 4, “Out of the 23 newborns” should be “23 infants”.
If the above questions could be answered and solved, the data will be more convincing.
Comments on the Quality of English LanguageThe English presentation is fine. Minor edting is required.
Author Response
Dear reviewer,
Thank you very much for these valuable comments.
Please see the attachment.
Sincerely,
Authors

Reviewer 2 Report
Comments and Suggestions for Authors
The authors' article is very interesting and well constructed but needs some minor but important revisions.
First, the range of infants aged from 28 days to one year of life is very broad and includes biological categories that cannot be grouped together.
For example: the problem of maternal lactation, with the possibility of transmitting the SARS-COV 2 virus or bacterial superinfections, or antibiotics and other therapeutic treatments and/or the immune response; the problem of lung maturation, the cut-off of six post-natal months is important; the problem of the maturation of the cardio-pulmonary circulation with the closure of the Botallo duct in the first months of life and possibly closure defects of the foramen ovale, which can influence the sedimentation of microbiological agents in different lung lobes.
A further important fact, which is just mentioned by the authors, is the possibility of feto-placental viral transmission, of which there is numerous evidence in the literature. This would lead to a further subdivision of newborns infected with Covid through maternal-fetal transmission, compared to primary post-natal infection (for example, psychomotor agitation may be linked to reduced maternal-fetal perfusion in mothers infected with Covid).
Small further clarification, the authors describe the B lines in lung ultrasound, without giving a pathophysiological explanation.
Author Response

(The authors gave the same response as above.)

Round 2
Reviewer 1 Report
Comments and Suggestions for Authors
The authors have answered most of the questions I mentioned. Even though not all the answers fulfilled my expectation, they were acceptable for the manuscript.
Comments on the Quality of English LanguageThe English presentation is fine. Minor edting is required.